# Synergic Action of Insulin-like Growth Factor-2 and miRNA-483 in Pterygium Pathogenesis

**DOI:** 10.3390/ijms24054329

**Published:** 2023-02-22

**Authors:** Cristina Maxia, Michela Isola, Eleonora Grecu, Alberto Cuccu, Alessandra Scano, Germano Orrù, Nick Di Girolamo, Andrea Diana, Daniela Murtas

**Affiliations:** 1Department of Biomedical Sciences, University of Cagliari, 09042 Monserrato, Italy; 2Department of Surgical Science, Eye Clinic, Azienda Ospedaliero-Universitaria (AOU), 09123 Cagliari, Italy; 3Department of Surgical Sciences, Molecular Biology Service Laboratory, University of Cagliari, 09123 Cagliari, Italy; 4Department of Pathology, School of Biomedical Sciences, Faculty of Medicine and Health, University of New South Wales, Sydney, NSW 4385, Australia

**Keywords:** pterygium, IGF-2, IGF-1R, miR-483, oxidative stress, *IGF2* LOI

## Abstract

Pterygium is a multifactorial disease in which UV-B is speculated to play a key role by inducing oxidative stress and phototoxic DNA damage. In search for candidate molecules that are useful for justifying the intense epithelial proliferation observed in pterygium, our attention has been focused on Insulin-like Growth Factor 2 (IGF-2), mainly detected in embryonic and fetal somatic tissues, which regulate metabolic and mitogenic functions. The binding between IGF-2 and its receptor Insulin-like Growth Factor 1 Receptor (IGF-1R) activates the PI3K-AKT pathway, which leads to the regulation of cell growth, differentiation, and the expression of specific genes. Since IGF2 is regulated by parental imprinting, in different human tumors, the IGF2 Loss of Imprinting (LOI) results in IGF-2- and IGF2-derived intronic miR-483 overexpression. Based on these activities, the purpose of this study was to investigate the overexpression of IGF-2, IGF-1R, and miR-483. Using an immunohistochemical approach, we demonstrated an intense colocalized epithelial overexpression of IGF-2 and IGF-1R in most pterygium samples (Fisher’s exact test, *p* = 0.021). RT-qPCR gene expression analysis confirmed IGF2 upregulation and demonstrated miR-483 expression in pterygium compared to normal conjunctiva (253.2-fold and 12.47-fold, respectively). Therefore, IGF-2/IGF-1R co-expression could suggest their interplay through the two different paracrine/autocrine IGF-2 routes for signaling transfer, which would activate the PI3K/AKT signaling pathway. In this scenario, miR-483 gene family transcription might synergically reinforce IGF-2 oncogenic function through its boosting pro-proliferative and antiapoptotic activity.

## 1. Introduction

Pterygium is a common and sporadic disease of the ocular surface, described as a wing-shaped or triangular overgrowth of conjunctival mucosa extending on the cornea. Possible pathological consequences encompass corneal astigmatism and the obstruction of the visual axis, often contributing to visual loss and diplopia [1]. It is a chronic, degenerative, and hyperplastic disorder in which epithelial proliferation, goblet cell hyperplasia, angiogenesis, inflammation, elastosis, stromal plaques, and Bowman’s membrane dissolution are common features. Despite being considered a benign lesion, pterygium has been suggested to be a neoplastic-like growth disorder for its tumor-like traits (aggressive recurrence after removal and local invasiveness) and association with preneoplastic lesions [2,3].

Notably, pterygium is referred to as an “ophthalmic enigma” [4] because it represents a multifactorial disease as a result of a multitude of risk factors, including inflammation [5,6,7], antiapoptotic growth and proliferative mechanisms [8,9], angiogenic factors [10,11], oxidative stress and hypoxic ischemic injuries [12,13], viral infections [14], and extracellular matrix remodeling [15]. However, there is scientific consensus in the perception of pterygium as one of the most common sun-related eye diseases (ophthalmohelioses) [16], with UV-B playing a key role through inducing oxidative stress and is responsible for phototoxic DNA damage [17,18].

In search for candidate molecules that are useful for justifying the intense epithelial proliferation observed in pterygium, our attention has been focused on Insulin-like Growth Factor 2 (IGF-2), also known as Somatomedin A, a single-chain polypeptide hormone belonging to the Insulin-like Growth Factors’ (IGFs) system, which regulates metabolic and mitogenic functions [19,20]. Unlike Insulin-like Growth Factor 1 (IGF-1), preferentially expressed postnatal IGF-2 is mainly detected in embryonic and fetal somatic tissues and in adult liver, meninges, and choroid plexuses [21], even though traces of IGF-2 can be identified in adult cerebral spinal fluid, possibly related to neurogenesis in the subventricular and subgranular zone of the brain [20].

IGF-2 can specifically bind to three distinct receptors, namely, Insulin-like Growth Factor 2 Receptor (IGF-2R), the isoform A of Insulin Receptors (IR-A), and Insulin-like Growth Factor 1 Receptor (IGF-1R). Moreover, it interacts with IGF-2R, a nonsignaling receptor, which acts as a scavenger for circulating IGF-2 and limits its bioavailability via lysosomal internalization and degradation [22]. Furthermore, IGF-2 is a natural ligand for IR-A with affinity properties related to the receptor close to the same insulin, and for IGF-1R, competing with IGF-1 [19,23,24]. IGF-2/IGF-1R binding activates the phosphatidylinositol 3-kinase (PI3K)-AKT/protein kinase B (PKB) pathway that leads to the regulation of cell growth, differentiation, and the expression of specific genes [25]. Indeed, the activation of the PI3K/AKT cascade has been recently demonstrated in pterygium [26]. In humans, the *IGF2* gene is localized on chromosome 11p15.5 and is parentally imprinted. The *IGF2* epigenetic deregulation known as “*Loss of Imprinting*” (LOI) results in IGF-2 overexpression, which has been associated with neoplastic progression [24] and in the overexpression of *IGF2*-derived intronic miR-483 [27] abnormally expressed in different tumors. Moreover, miR-483 overexpression has been correlated with the promotion of cell proliferation in colorectal cancer [28] and in apoptosis inhibition in hepatocellular carcinoma [29]. Notably, in some benign and malignant tumors, *IGF2* expression is directly regulated by *Pleomorphic adenoma gene 1 (PLAG1)* [30], inducing autocrine IGF-1R signaling that leads to the activation of PI3K/AKT downstream cascade, enhancing cell survival and proliferation [31].

Therefore, the main purpose of this study was to detect the coexistence of IGF-2, IGF-1R, and miR-483 and their synergic action in pterygium, aiming to support the proposition that considers pterygium a benign lesion, wherein the uncontrolled epithelial cell proliferation is the result of autocrine/paracrine cell activation and dysregulation of apoptosis.

## 2. Results

### 2.1. IGF-2 Immunoistochemical (IHC) Expression

The results for IGF-2 IHC expression in pterygium and conjunctiva are illustrated in Figure 1. In our study, 45 (45/48, 93.6%) primary pterygium samples showed IGF-2 immunoreactivity. The staining was localized only in the cytoplasm of epithelial cells, and immunoreactivity was not detected in any nuclei (Figure 2A). In detail, in all samples, the staining was detectable in the basal and suprabasal layers of the epithelium, as well as in nine (9/45, 20%) pterygium specimens also in the superficial cells. In nine (9/11, 81.8%) of the normal conjunctiva samples, the immunostaining was absent (Figure 2B). No staining was shown in the negative control (Figure 2C).

### 2.2. IGF-1R IHC Expression

The results for IGF-1R IHC expression in pterygium and conjunctiva are represented in Figure 1. Thirty-four (34/48, 70.8%) pterygia exhibited IGF-1R immunoreactivity in the cytoplasm of epithelial cells located in the basal and middle layers, and, in 17 (17/34, 50%) of these samples, the staining covered a larger area in the superficial layers of the epithelium (Figure 3A). In most normal conjunctiva samples (10/11, 90.9%), the immunostaining for IGF-1R was not discernible (Figure 3B). No staining was present when the primary antibody was omitted or when an isotype control antibody was applied (Figure 3C).

### 2.3. Relationship between IGF-2 and IGF-1R Expression

The relationship between IGF-2 and IGF-1R is shown in Table 1. In the group of pterygia with IGF-2-positive immunostaining, there were 34 (34/45, 75.6%) samples with IGF-1R expression, while the remaining 11 (11/45, 24.4%) specimens did not show any IGF-1R immunoreactivity. Moreover, only three (3/48, 6.3%) samples were negative for both IGF-2 and IGF-1R, and there were no IGF-2^-^/IGF-1R^+^ samples. A significant correlation between IGF-2 expression and IGF-1R staining in primary pterygium was demonstrated using Fisher’s exact test (*p* = 0.021) (Table 1), further confirmed through double immunofluorescence (IF) staining (Figure 4).

### 2.4. Molecular Analysis

All the recruited samples for the molecular analysis showed sufficient RNA quantity (>0.5 ng/mL) for the evaluation of the genetic expression. Gene sequences referring to *IGF2* and miR-483 were both overexpressed in pterygium compared with normal conjunctiva patients by 253.2 and 12.47 folds (respectively), assuming that, in healthy subjects, the expression rate of the test gene/housekeeping gene is considered = 1 (Figure 5).

## 3. Discussion

Pterygium is often described as a multifactorial disease with UV-rays thought to play a key role in its pathogenesis. With the aim of disclosing information about the factors possibly involved in the occurrence and progression of pterygium, we evaluated the presence of IGF-2, a well-known factor implicated in embryonic development and tumorigenesis, due to its role in regulating cell proliferation, growth, migration, differentiation, and survival as a result of its paracrine/autocrine effect [32]. As a matter of fact, alterations in the autocrine loops IGF-2/IGF-1R often occur in human cancer, leading to the overactivation of the PI3K/AKT signaling pathway, which is responsible for impairment in cell proliferation and apoptosis [20]. Since IGF-2 dysregulation has been already shown in childhood and adult malignancies [33], as well as in benign tumors [34,35,36], it seemed reasonable to investigate the involvement of the IGF-2/IGF-1R pathway in primary pterygium, a benign tumor with premalignant features. In our study, the specific IHC overexpression ofIGF-2 was observed in 93.6% of the examined pterygium samples, with the immunostaining detected in the cytoplasm of epithelial cells being localized in the basal and suprabasal layers; moreover, in nine (9/45, 20%) samples, the IGF-2-positive cells spread across the superficial layers of the epithelium. Only 18.2% of the normal conjunctiva specimens displayed immunostaining for IGF-2. These results were further confirmed by our molecular analysis. IGF-2 upregulation in pterygium might be explained by a possible occurrence of the IGF-2 epigenetic dysregulation, known as “loss of imprinting”, during aging. Indeed, Yang et al. [37] suggested that the activation of the canonical Nuclear Factor Kappa-light-chain-enhancer of activated B cells’ (NF-κB) signaling pathway, induced by oxidative stress, is responsible for *IGF2* LOI. In addition, NF-κB acts as a transcription factor for Hypoxia Inducible Factor 1 Subunit Alpha (*HIF1A*) [38], which, in turn, translocates into the nucleus and targets the *IGF2* gene, resulting in enhanced cell proliferation. Since UVB exposure, responsible for irritative stimuli, induces oxidative stress-mediated NF-κB activation [39,40], it is definitely consistent with the notion that LOI-dependant IGF-2 overexpression may contribute to pterygium development. Likewise, in colorectal cancer tissues, it has been demonstrated that the hypomethylation of a “differently methylated region” (DMR), located adjacent to *IGF2*, might occur concurrently with *IGF2* LOI, contributing to IGF-2 upregulation [41].

Several studies have indicated that *IGF2* LOI drives the concomitant overexpression of the miR-483 family gene, embedded in the second intron of *IGF2*, which reinforces IGF-2 oncogenic function [42]. Since the upregulation of miR-483 has been associated to IGF-2 presence, the molecular analysis of qRT-PCR identified a well-defined *IGF2* and miR-483 co-overexpression in pterygium in comparison with normal conjunctiva; in detail, the mean fold-increase expression values for *IGF2* and miR-483 were 253.2 and 12.47, respectively. A possible explanation for the different expression levels between *IGF2* and miR-483 might be offered by the genetic nature of the miR-483 family itself. Indeed, while hepatocellular carcinoma in miR-483 has been demonstrated to be co-expressed with its host gene *IGF2* [43], in pterygium, there is a lack of knowledge about this interaction. miR-483 splicing produces miR-483-5p and -3p, which are processed from the same stem-loop precursor miRNA (pre-miR-483) encoded within the intron of the *IGF2* locus. Interestingly, miR-483-5p has been shown to bind directly to the 5′ untranslated region (UTR) of the *IGF2* primary transcript, thereby upregulating the expression of its host gene [44] and exerting its pro-proliferative effect. This peculiar miR-483-5p behavior might appear to be surprising, since miRNAs generally act by inhibiting their target gene. However, recent studies have shown that several miRNAs located in the nucleus can alternate into both the silent and activated status within the gene transcription [45]. Hence, this positive feedback loop might justify the more evident *IGF2* expression than that observed for miR-483. Furthermore, in liver cancer [44], miR-483-3p has been found to act as an oncogenic and antiapoptotic factor by means of the inhibition of the p53 Upregulated Modulator of Apoptosis (PUMA), a pro-apoptotic protein that, under physiological conditions, blocks the antiapoptotic action of B-Cell CLL/Lymphoma 2 (BCL2) [46]. As a matter of fact, this hypothesis in pterygium has been supported by the finding of BCL-2 overexpression [47,48]. Based on the above evidence, we reason that the presence of miR-483 might be consistent with the uncontrolled epithelial cell proliferation and dysregulation of apoptosis associated to pterygium. However, further studies are required to verify the presence of both miRNAs (miR-483-3p, miR-483-5p), which would support the oncogenic function of IGF-2. 

The importance of the IGF axis in tumors was unambiguously claimed following the observation that embryonic fibroblasts lacking the IGF-1R are not susceptible of transformation [49]. In our study, 70.8% primary pterygia exhibited IGF-1R immunoreactivity in the cytoplasm of epithelial cells located in the basal and middle layers; moreover, in 17 (17/34, 50%) of these samples, the staining was enlarged to the superficial layers of the epithelium. Immunostaining for IGF-1R in normal conjunctiva samples was detected only in one (1/11, 9.09%) sample. Fisher’s exact test demonstrated a significant correlation between IGF-2 and IGF-1R overexpression (*p* = 0.021), since, within the group of 45 IGF-2^+^ samples, 34 (34/45, 75.6%) were also IGF-1R^+^, while the remaining 11 (11/45, 24.4%) specimens were IGF-1R^-^. As demonstrated by the results of the double IF analysis, IGF-2 and IGF-1R were co-expressed in the epithelial cells of the basal and middle layers. Therefore, we envision that IGF-2/IGF-1R co-expression in most pterygium samples would favor their interplay through the two different paracrine/autocrine IGF-2 routes for signaling transduction. The paracrine modality would activate the PI3K/AKT pathway, carrying out the downstream transcription of *HIF1A*, whose protein is a crucial factor involved in the occurrence and development of pterygium [11]. As a confirmation of that, HIF-1a stimulates the transcription of *IGF2* and Vascular Endothelial Growth Factor A (*VEGFA*) genes [50] as a tumoral adaptive response to hypoxia [51]. The autocrine pathway would constantly supply the PI3K/AKT signal cascade, contributing to IGF-2 overexpression through a positive feedback loop mediated by PLAG1, as demonstrated in some benign and malignant tumors [30].

However, the finding of 11 IGF-2^+^/IGF-1R^-^ samples led us to speculate that the IGF-1R-mediated PI3K/AKT pathway would not be the unique or the main pathway activated in pterygium. Indeed, according to Somers et al. [52], the numerical preponderance of the IGF-2-positive samples versus IGF-1R might arise from the existence of a subcellular localization of cytoplasmic IGF-2, whose function has not been well understood.

Hence, in our data interpretation (Figure 6), the IGF-2/IGF-1R co-expression in most pterygium samples implies their interplay through the two different paracrine/autocrine IGF-2 routes for signaling transfer, activating the PI3K/AKT signaling pathway, which is probably responsible for the increased epithelial cell proliferation. In this scenario, the miR-483 gene family transcription might synergically reinforce IGF-2 oncogenic function through its pro-proliferative and antiapoptotic activity. Moreover, considering the well-known tangled crosstalk between oxidative stress and hypoxia, the take-home message of the present study might be the existence, in pterygium, of a sort of feed-forward loop started by UV-induced oxidative stress that would activate NF-κB, also stimulated by the IGF-2/IGF-1R-induced PI3K/AKT pathway; this cascade would carry out the downstream transcription of *HIF1A*, a hypoxic factor involved in the occurrence and development of pterygium, which, in turn, would trigger the expression of IGF-2, and would be no more silenced as a result of the NF-κB-induced LOI (Figure 6). Therefore, even if the molecular mechanisms involved in pterygium pathogenesis are still partly unknown, the results of this study would further support the hypothesis of pterygium as a neoplastic-like multifactorial disease, in which a network of oncogenic pathways join and mutually boost each other. Moreover, based on the findings that IGF-2 and miR-483 are a promising translational research area for tumor therapy, the pharmacological inhibition of mitogenic signaling by targeting these molecules could be a challenge for pterygium treatment.

## 4. Materials and Methods

### 4.1. Patients and Study Design

The study was performed on a group of 52 primary pterygium samples obtained from 42 male and 10 female patients, ranging in age from 36 to 80 years (mean 56.5 yrs years), who underwent surgery for pterygium removal at the Department of Surgical Science, Eye Clinic of the University of Cagliari. Most pterygia were located on the nasal side (33, 63%), and their morphology was clinically graded as atrophic (n = 14, 27%), intermediate (n = 29, 56%), or fleshy (n = 9, 17%) by means of pterygium translucency assessment, which included n = 33 (63%) inflamed and n = 19 (37%) quiescent lesions. Moreover, 15 normal epibulbar conjunctiva samples, used as control tissues, were excised from healthy donors (9 males and 6 females, ranging in age from 13 to 77 years), with no signs or symptoms of pterygium or a conjunctival disorder during strabismus or cataract surgery.

The study was approved by the Independent Ethic Committee (CEI) of the Azienda Ospedaliero-Universitaria (AOU), Cagliari (Prot. NP/2022/5126), and written consent was obtained from all patients before the beginning of the study, in accordance with the World Medical Association Declaration of Helsinki. Patients did not receive any medication before surgery except for topical anesthetic; demographic and clinical characteristics of patients, available in all cases, are reported in Table 2.

### 4.2. Immunohistochemical (IHC) Analysis

Among the 52 pterygium samples, 48 specimens were 10% formalin-fixed and paraffin-embedded; moreover, 11 epibulbar conjunctiva specimens, belonging to a group of 15 samples from healthy donors, were formalin-fixed, processed for paraffin embedding, and used as normal controls. Three microtome histological sections (6–7 µm thick) per sample were subjected to immunohistochemistry (IHC) for the demonstration of IGF-2 and IGF-1R antigens using the streptavidin-biotin alkaline phosphatase method, as previously described [7,53]. These were dewaxed, rehydrated, and rinsed in phosphate-buffered saline (PBS), pH 7.4. A heat-induced IGF-2 epitope retrieval (HIER) procedure was carried out via the immersion of the samples in a water bath-heated TRIS/EDTA buffer (TRIS 10 mM + EDTA 1 mM, pH 9.0) for 30 min at 95 °C, followed by gradual cooling for 20 min at room temperature (RT); conversely, IGF-1R epitope did not require any antigenic retrieval treatment. Furthermore, sections were treated for 45 min at RT with 10% nonimmune serum to block nonspecific binding and then incubated with the primary antibodies 1 h at RT. Table 3 includes reported sources, dilutions, time of incubation, and the details of the primary antibodies used for the IHC staining, according to the Resource Identification Initiative [54]. 

Biotinylated anti-mouse or anti-rabbit secondary antisera (1:200; Vector Laboratories, Burlingame, CA, USA) were incubated for 30 min at RT; then, the sections were treated with alkaline-phosphatase streptavidin (1:1000, Vector Laboratories, Burlingame, CA, USA) for 30 min at RT and were reacted using the SIGMA FAST^TM^ Fast red substrate–chromogen system (SIGMA, St. Louis, MO, USA). All sections were carefully rinsed in PBS after each step and were finally counterstained using Carazzi haematoxylin and mounted in glycerol gelatin (SIGMA, St. Louis, MO, USA). 

Positive and negative controls were run simultaneously to evaluate reaction specificity. Archival sections of normal human skin and human seminoma were used as known positive controls for IGF-2 and IGF-1R, respectively. Negative controls were carried out by omitting the primary antibody or by replacing the primary antibody with an isotype-matched antibody. 

A Zeiss Axioplan2 microscope (Carl Zeiss Vision, Hallbergmoos, Germany) equipped with the following objectives: 20×/0.45 Zeiss Achroplan; 40×/0.75 Zeiss Plan-Neofluar, 63×/1.40 oil immersion Zeiss Plan-Apochromat, was used for the analysis of immunolabelled slides, while image capture was performed using a Lumenera Infinity 3-1URC camera (1.4 megapixels; Lumenera Corporation, Ontario, Canada) and Infinity Capture 6.3.0 software (Lumenera Corporation, Ontario, Canada). The figure panels’ creation and a slight adjustment of brightness and contrast were carried out using Adobe Photoshop CS3 Extended (ver. 10.0, Adobe Systems Incorporated, CA, USA).

### 4.3. Scoring

Three experienced observers (CM, EG, and DM) independently evaluated the immunoreactivity in a triple-blind fashion, as previously described [18]. In each of the three sections per sample, the percentage of epithelial immunoreactive cells was scored in four to six randomly chosen microscopic fields (×40 original magnification), covering almost the whole field of each section per sample. Furthermore, in all samples, epithelial cellular compartments (nucleus, cytoplasm, or both) were analyzed for the localization of the immunoreactivity. IGF-2 and IGF-1R cut-off was set at 10%, which accounts for the percentage of stained epithelial cells per positive sample. Samples were divided into four groups, i.e., IGF-2-positive, IGF-2-negative, IGF-1R-positive, and IGF-1R-negative.

### 4.4. Double Immunofluorescence (IF)

A simultaneous procedure was used for the double staining of IGF-2 and IGF-1R, as previously described [55]. Following deparaffinization, rehydration, antigen retrieval, and blocking of nonspecific binding, sections of pterygium and conjunctiva were incubated for 1 h at RT with a mixture of mouse monoclonal anti-human IGF-2 (clone 8H1, 1:100, ThermoFisher Scientific, Waltham, MA USA 02451) and rabbit polyclonal anti-human IGF-1R (1:50; Abcam Cambridge, CB2 0AX, UK) primary antibodies. Alexa Fluor 594 donkey anti-mouse IgG (H+L) and Alexa Fluor 488 donkey anti-rabbit IgG (H+L) (1:200, Invitrogen Life Technologies, Paisley, UK) secondary antibodies were used for immunofluorescence detection. The sections were mounted in VECTASHIELD^®^ Antifade Mounting Medium with 4′,6-diamidino-2-phenylindole (DAPI) (Vector Laboratories) to visualize nuclear details. Immunofluorescence-labelled sections were imaged using a Zeiss Axioplan2 microscope (HBO 100 illuminator; mercury vapor, short arc lamp) equipped with the appropriate filter sets to distinguish the chosen fluorochromes and the above-mentioned objectives and digital camera. The displayed figure panel was set up using Adobe Photoshop.

### 4.5. Statistical Analysis

The immunohistochemical results were analyzed using Fisher’s exact test. Data were computed with IBM^®^ SPSS^®^ Statistics 21.0. The tests used were two-tailed. A *p* value ≤ 0.05 was considered statistically significant.

### 4.6. Molecular Analysis

#### 4.6.1. RNA Extraction 

Among the group of 52 pterygium and 15 conjunctiva samples, 4 pterygium and 4 epibulbar conjunctiva specimens, respectively, were addressed in the molecular analysis. Immediately after the removal, each sample was placed inside an Eppendorf^®^ 2-mL tube containing 100 µL of RNAlater^®^ solution and then frozen at −80 °C until RNA extraction.

The samples stored in the RNAlater^®^ solution were thawed and then used for RNA extraction by using a RNeasy^®^ mini-Kit (QIAGEN GmbH, Hilden, Germany), following the manufacturer’s instruction. The quality of the RNA extract was evaluated using a NanoDrop^®^ Instrument (ThermoFisher Scientific, Waltham, MA USA), according to the manufacturer’s directions. The optimal requirements for the A260/A280 ratios were 1.8–2.2, while the requirements for the A260/A230 ratios were >1.7. Regarding the concentration required for Real-Time quantitative PCR (RT-qPCR) amplification (threshold cycle CT > 35), the samples showed a concentration > 0.5 ng/mL, and the RNAs were resuspended in 30 μL of nuclease-free H_2_O and kept at −80 °C until use.

#### 4.6.2. Gene Expression

Relative expression of miR483 and IGF2 genes was performed through an RT-qPCR procedure using the 2^−∆∆Ct^ method [56]. To avoid oligonucleotide dimers formation and self-complementarity, and to set the correct annealing temperatures of the RT-qPCR, the oligos were designed using the Oligo program vers. 6 (MedProbe, Oslo, Norway) and “UNAFold Web Server” programs (http://www.unafold.org/mfold/applications/dna-folding-form.php) (accessed on 10 October 2022) using the procedures previously described [57]. The theoretic melting temperatures of the different PCR amplicons (Tm_s_) were calculated using module 2 of the UNAFold Web Server. Oligonucleotide sequences and their thermodynamic parameters are described in Table 4; all primer pairs were checked for their efficacy on the serial dilutions of cDNA. The RT-PCRs showed an efficiency range from 0.95 to 0.98 [58].

#### 4.6.3. Quantitative RT-PCR Procedure

Retro-transcription was carried out by using the SuperScript^®^ VILO™ cDNA Synthesis Kit (Invitrogen, ThermoFisher Scientific, Waltham, MA USA), following the manufacturer’s instructions. RT-qPCR was performed in a Light Cycler apparatus (Roche, Basel, Switzerland) by using the LightCycler^®^ FastStart DNA Master HybProbe Kit (Roche Molecular Systems, Inc., Rotkreuz, Switzerland). In 20 µL of the PCR reaction mix, a mixture of 2 µL of cDNA extract, 5 pmol of each primer, 10 pmol of TaqMan probe, and 2 µL of SYBR Green solution (Table 4) was present. The PCR amplification conditions consisted of (i) 30 s (s) of denaturation at 95 °C; (ii) 40 cycles of 1 s at 95 °C, 10 s at 52 °C, and 8 s at 72 °C. The temperature transition rate in the denaturation and annealing steps was 20 °C/s, while, during the polymerization step, it was 5 °C/s. Fluorescence was recorded at a wavelength between 510 nm and 550 nm at the end of each PCR cycle. Three distinct biological replicas were obtained, and quantitative data were expressed as mean ± SD. Changes in gene expression above 2 or below 0.5 were considered significant.

## Figures and Tables

**Figure 1 ijms-24-04329-f001:**
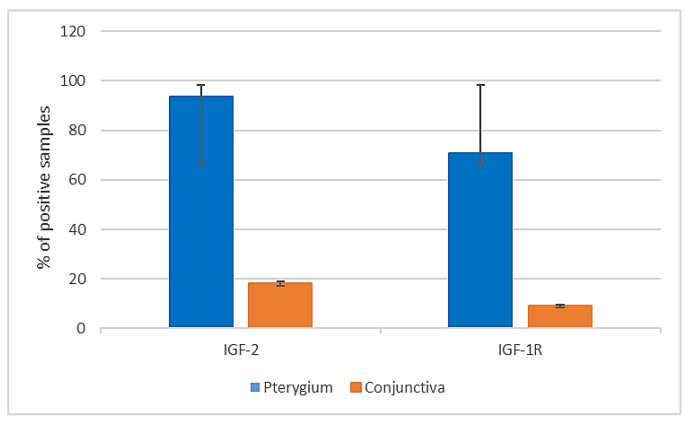
IGF-2 and IGF-1R IHC expression in pterygium and conjunctiva, expressed as % of immunoreactive samples. A total of 48 pterygium and 11 conjunctiva samples were analyzed. The bars represent the standard deviation.

**Figure 2 ijms-24-04329-f002:**
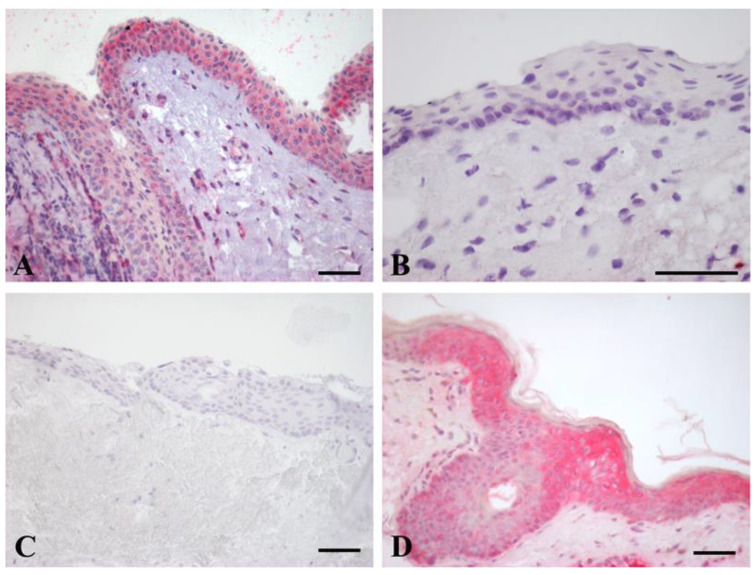
IGF-2 IHC expression: (**A**) pterygium; (**B**) conjunctiva; (**C**) pterygium, negative control (**D**) human skin, positive control. Counterstained with Carazzi hematoxylin. (**A**–**D**) Scale bar = 25 µm.

**Figure 3 ijms-24-04329-f003:**
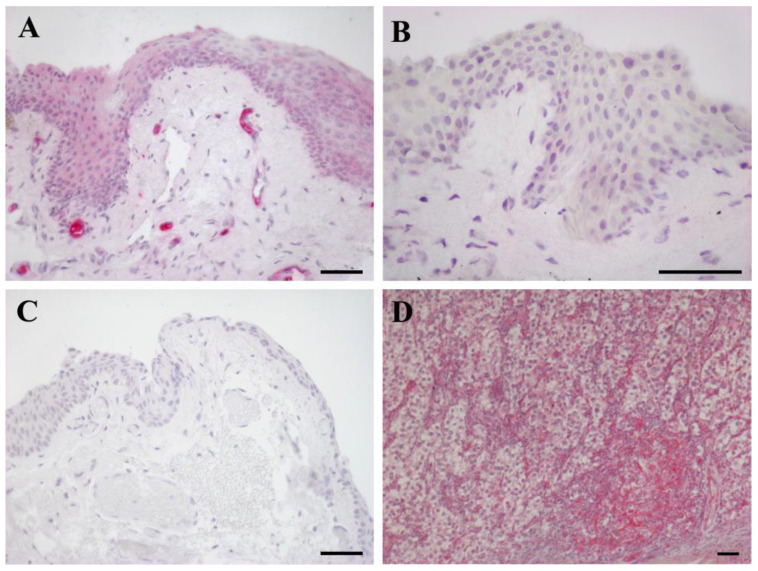
IGF-1R IHC expression: (**A**) pterygium; (**B**) conjunctiva; (**C**) pterygium, negative control (**D**) human seminoma, positive control. Counterstained with Carazzi hematoxylin. (**A**–**D**) Scale bar = 25 µm.

**Figure 4 ijms-24-04329-f004:**
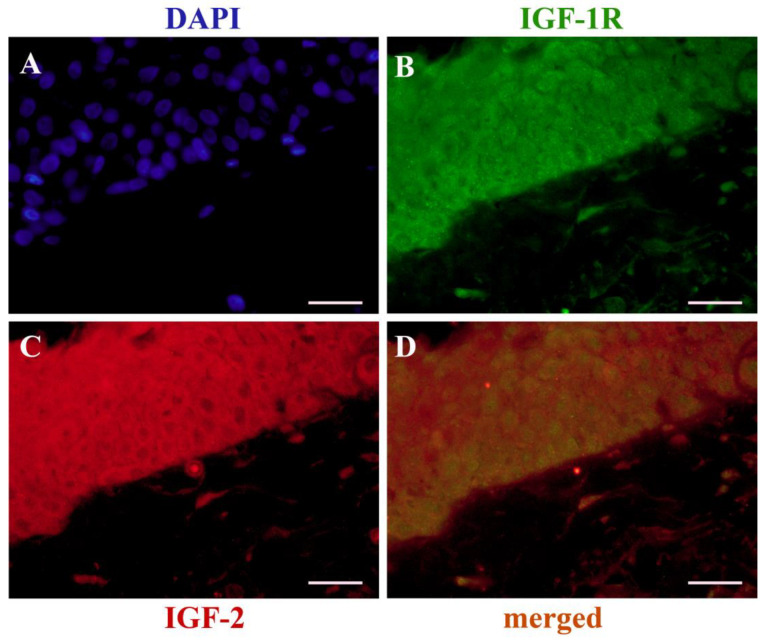
Immunofluorescence for IGF-2/IGF-1R in pterygium: (**A**) nuclei counterstained with DAPI (blue); (**B**) anti-human IGF-1R (green); (**C**) anti-human IGF-2 (red); (**D**) expression (yellow/orange) of IGF-2 and IGF-1R. (**A**–**D**): Scale bar = 20 µm.

**Figure 5 ijms-24-04329-f005:**
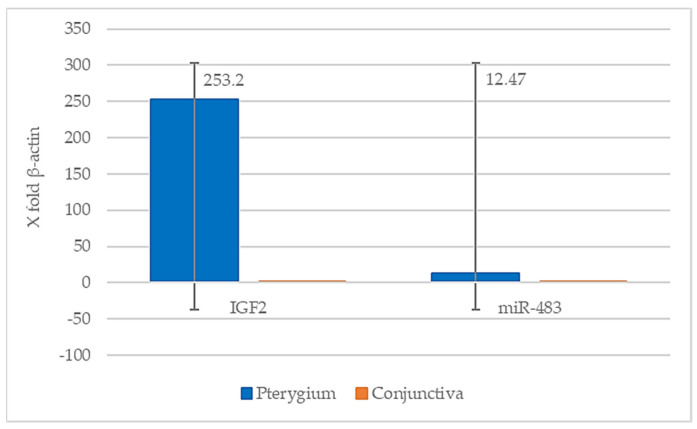
Mean value for *IGF2* and miR-483 gene expression patterns in pterygium and normal conjunctiva. The bars represent the standard deviation.

**Figure 6 ijms-24-04329-f006:**
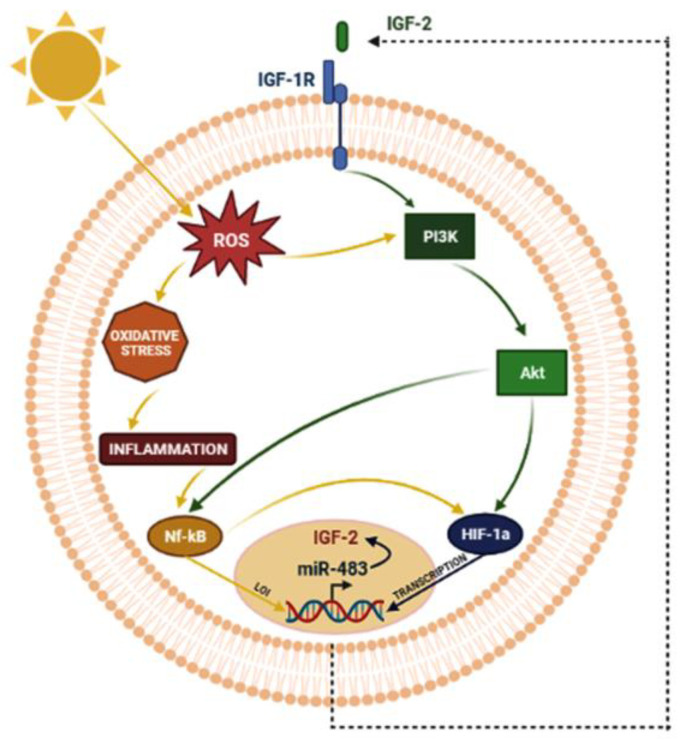
Possible crosstalk between UV-induced oxidative stress and hypoxia in pterygium (BioRender.com: https://biorender.com/ (accessed on 24 December 2022)).

**Table 1 ijms-24-04329-t001:** IGF-2 and IGF-1R relationship in pterygium (Fisher’s exact test).

	IGF-1R^+^	IGF-1R^-^	Total
IGF-2^+^	34	11	45
IGF-2^-^	0	3	3
Total	34	14	48
			*p* = 0.021

**Table 2 ijms-24-04329-t002:** Clinical features of patients.

Patients’ Information		Pterygium (n. 52)	Conjunctiva (n. 15)
Mean age (yrs)		56.5	38.87
Age range (yrs)		36–80	13–77
Sex (%)	Male	42 (81%)	9 (60%)
	Female	10 (19%)	6 (40%)
Location of the lesion (%)	Nasal side	33 (63%)	
	Temporal side	19 (37%)	
Grade (%)	Atrophic	14 (27%)	
	Intermediate	29 (56%)	
	Fleshy	9 (17%)	
Disease stage (%)	Inflamed	33 (63%)	
	Quiescent	19 (37%)	

**Table 3 ijms-24-04329-t003:** Characteristics of the primary and secondary antibodies.

Target	Primary Antibody						Biotinylated Secondary Antibody
	Vendor	Origin	Dilution	Incubation	Antigen Retrieval	RRID *	
IGF-2	Thermo Fisher Scientific	Mouse (mc) (clone 8H1)	1:200	1 h RT	HIER	AB_2538567	Horse anti-mouse IgG ^1^
IGF-1R	Abcam	Rabbit (pc)	1:100	1 h RT	n.r.	AB_731541	Goat anti-rabbit IgG ^1^

Abbreviations: mc, monoclonal; pc, polyclonal. RT, room temperature. n.r., not required. HIER, heat-induced epitope retrieval at 95 °C in TRIS 10 mM + EDTA 1mM (pH 9.0) for 30 min; ^1^ Vector Laboratories, Burlingame, CA, USA, dilution 1:200 for 30 min RT. * The Resource Identification Initiative [54]; Resource Identification Portal (https://scicrunch.org/resources (accessed on 24 December 2022)).

**Table 4 ijms-24-04329-t004:** Oligonucleotide sequences for primers and probes utilized for gene expression.

Oligo/Probe Name	Sequence 5′ → 3′	Position	Length/Tm °C
miR-483 gene expression ^a^			
OG 686 Mir-483-F	GAGGGGGAAGACGGGAGG	32001	18/54.0
OG 687 Mir-483-R	GAGAGGAGAAGACGGGAGGA	32057	20/50.7
OG 688 TaqMan Mir-483 Probe	Fam-GGCGTGATGGAACCACTCCCT-BHQ-1	32024	21/58.3
IGF-2 gene expression ^a^			
OG689 (IGF-2) F	GGACAACTTCCCCAGATACCCC	1696	22/56
OG690 (IGF-2) R	TCACTTCCGATTGCTGGCCA	1914	20/57.2
SYBR Green Probe			
Housekeeping gene (β-actin) ^b^			
OG650(Beta-act) F	GCATGGGTCAGAAGG	221	15/51.8
OG651(Beta-act) R	AGGCGTACAGGGATAG	502	16/51.2
SYBR Green Probe			

Legend: referred to GenBank sequences accession number NG 050578.1 ^a^ and NM 001101 ^b^.

## Data Availability

The data that support the findings of this study are available upon request.

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
