# Peer review of "Synergic Action of Insulin-like Growth Factor-2 and miRNA-483 in Pterygium Pathogenesis"

_ijms, 2023, doi:10.3390/ijms24054329_

Round 1

Reviewer 1 Report

The authors investigate the overexpression of Insulin-like Growth Factor 2 (IGF-2), Insulin-like Growth Factor 1 receptor (IGF-2)IGF-1R and miR-483 in Pterygium, a multifactorial disease that shows an intense epithelial proliferation.

Their attention has been focused on IGF-2, because detected in embryonic and fetal somatic tissues, and is also a regulator of some metabolic and mitogenic functions. In particular, the IGF-2 binding with IGF-1R activates the PI3K-AKT pathway, leading to the regulation of cell growth and differentiation.

They used different techniques as the immunohistochemical approach and RT-qPCR gene expression analysis on different primary pterygium samples obtained from 42 male and 10 female patients.

The authors suggest that IGF-2/IGF-1R colocalization in most pterygium samples would mean their interplay through different routes for signalling transfer, activating the PI3K/AKT signalling pathway (increasing epithelial cell proliferation). Moreover, miR-483 family gene transcription might reinforce IGF-2 oncogenic function, through its pro-proliferative and anti-apoptotic activity. The paper is interesting, and the experimental methods are clear, but some minor revisions are necessary.

Revisions:

1)    Page 2, lines 59-61:
“IGF-2 can specifically bind to three distinct receptors, interacting with Insulin-like Growth Factor 2 Receptor (IGF-2R), a non-signaling receptor acting as a scavenger for circulating IGF-2 and limiting its bioavailability by lisosomial internalization and degradation [22].”
Which receptors? Does it describe only the IGF-2R.
Please rewrite this paragraph to explain it better.

2)    In Figure 3: It is written: ".... Nuclei (pale blue) are counterstained with 131 4′,6-diamidino-2-phenylindole (DAPI)."

However, in immunofluorescence images, it is impossible to perceive any kind of blue signal.

3)    Graphic 1: Please enter the Y-axis title to explain the histogram values, How many samples were analyzed?

4)    In Graph 1 and Graph 2: It is written: “The bars represent the standard percentage error (10%).”

Why did the authors not use the standard deviation?

Author Response

ANSWERS TO REVIEWER 1

1)    Page 2, lines 59-61:

“IGF-2 can specifically bind to three distinct receptors, interacting with Insulin-like Growth Factor 2 Receptor (IGF-2R), a non-signaling receptor acting as a scavenger for circulating IGF-2 and limiting its bioavailability by lisosomial internalization and degradation [22].”

Which receptors? Does it describe only the IGF-2R. Please rewrite this paragraph to explain it better.”

We thank Reviewer 1 for the suggestion and, taking it into account, we modified the paragraph (lines 71-77) of the revised manuscript as follows:

“IGF-2 can specifically bind to three distinct receptors: Insulin-like Growth Factor 2 Receptor (IGF-2R), the isoform A of Insulin Receptors (IR-A), and Insulin-like Growth Factor 1 Receptor (IGF-1R). It interacts with IGF-2R, a non-signaling receptor which acts as a scavenger for circulating IGF-2 and limiting its bioavailability by lysosomal internalization and degradation [22]. Furthermore, IGF-2 is a natural ligand for IR-A, with affinity properties close to the same Insulin, and for IGF-1R, competing with IGF-1 [19,23,24]. IGF-2/IGF-1R binding activates the phosphatidylinositol 3-kinase (PI3K)-AKT/protein kinase B (PKB) pathway that leads to the regulation of cell growth, differentiation, and the expression of specific genes [25].

“2)    In Figure 3: It is written: ".... Nuclei (pale blue) are counterstained with 4′,6-diamidino-2-phenylindole (DAPI)."

However, in immunofluorescence images, it is impossible to perceive any kind of blue signal.”

We are grateful to Reviewer 1 for the observation. We totally agree with him/her and modified Figure 3 by adding a separate picture of nuclei counterstained with DAPI. Furthermore, we also modified the relative figure legend accordingly.

“3)    Graphic 1: Please enter the Y-axis title to explain the histogram values, How many samples were analyzed?”

Following the suggestion of Reviewer 1, we entered Y-axis title in Graphic 1 and modified the relative legend by adding the total number of pterygium and conjunctiva samples that were analyzed.

“4)    In Graph 1 and Graph 2: It is written: “The bars represent the standard percentage error (10%).”

Why did the authors not use the standard deviation?”

We totally agree with Reviewer 1 for the observation, and we apologize for the incorrect information reported in Graphic 1 and 2. We modified both Graphics and relative legends by indicating the standard deviation, instead of the standard percentage error.

Reviewer 2 Report

Cristina et al. present  Synergic Action of Insulin-like Growth Factor-2 and miRNA-483 in Pterygium Pathogenesis. The comparative novelty of this work is to evaluate for the first time indicating to detect the coexistence of IGF-2, IGF-1R, miR-483 and their synergic action in pterygium, aiming to support the hypothesis that considers pterygium as a benign lesion. 

In my opinion, this work is of interest to researchers in the field of a promising approach to the uncontrolled epithelial cell proliferation is the result of autocrine paracrine cell activation and dysregulation of apoptosis. The authors should consider the following comments to improve their manuscript. 

MA1. Graphic 1should be replaced to Figure 1. And it should be described in more details.

MA2. The Introduction must be improved by incorporating more recent references including action of Insulin-like Growth Factor-2 and miRNA-483 in Pterygium Pathogenesis.

MA3. Figures 1 and 2 should be merged and described in more detail in revised manuscript.

MA4.  Please address more details of synergistic action of Insulin-like Growth Factor-2 and miRNA-483 in Pterygium Pathogenesis in revised manuscript.

MA5. In conclusion, please the contents detailed should be addressed including future scope and applications for better understanding of the strength of synergistic action of Insulin-like Growth Factor-2 and miRNA-483 in Pterygium Pathogenesis and future perspectives.

The subject may be interesting enough ijms but only after major, deep revision, if at all possible, to resolve the above.

Author Response

ANSWERS TO REVIEWER 2

“MA1. Graphic 1 should be replaced to Figure 1. And it should be described in more details.”

Regarding to the order by which Graphic 1 and Figure 1 are included in the manuscript, we consider more appropriate to anticipate the semiquantitative evaluation of IGF-2 and IGF-1R, followed by the relative microscopic images, to give the reader an overall representation of these proteins’ expression. Based on these remarks, by purpose we showed first Graphic 1 and then Figure 1.

Figure 1 legend has been supplemented with more details, as also suggested by Reviewer 1.

“MA2. The Introduction must be improved by incorporating more recent references including action of Insulin-like Growth Factor-2 and miRNA-483 in Pterygium Pathogenesis.”

The Reviewer mentions at the beginning of his/her comment, “The comparative novelty of this work is to evaluate for the first time indicating to detect the coexistence of IGF-2, IGF-1R, miR-483 and their synergic action in pterygium, aiming to support the hypothesis that considers pterygium as a benign lesion”.

Indeed, we experienced a lack of findings on the action of IGF-2 and miRNA-483 in pterygium pathogenesis. Because of that, we aimed to explore this particular mechanism in pterygium.

Inspired by the Reviewer’ suggestions, we added the following sentence at line 241 of the revised manuscript: “in pterygium there is a lack of knowledge about this interaction.”

“MA3. Figures 1 and 2 should be merged and described in more detail in revised manuscript.”

We thank the Reviewer for the precise proposal, but we would prefer to keep the original setting of Figure 1 and 2 for a clearer and smoother description of the results. Furthermore, coherently with the Instructions to Authors, Figures legends have been arranged in the simplest fashion. However, Figure 1 legend has been supplemented with more details, as also suggested by Reviewer 1.

“MA4. Please address more details of synergistic action of Insulin-like Growth Factor-2 and miRNA-483 in Pterygium Pathogenesis in revised manuscript.”

We thank the Reviewer for the precise proposal; as a matter of fact, the synergic action of IGF-2 and miRNA-483 in pterygium has been described in Figure 4 and, in more detail, at lines 306-308 of the revised manuscript: “In this scenario, miR-483 gene family transcription might reinforce IGF-2 oncogenic function, through its pro-proliferative and anti-apoptotic activity……”. By the way, according to the useful Reviewer’ suggestion, to better emphasize our interpretation of the results, we added the word “synergically” at line 307 of the revised manuscript as follows: “miR-483 gene family transcription might synergically reinforce IGF-2 oncogenic function”.

“MA5. In conclusion, please the contents detailed should be addressed including future scope and applications for better understanding of the strength of synergistic action of Insulin-like Growth Factor-2 and miRNA-483 in Pterygium Pathogenesis and future perspectives.”

We thank the Reviewer for the useful suggestion. We added the following sentence at the end of the revised Discussion (lines 318-321): “Moreover, based on the findings that IGF-2 and miR-483 are a promising translational research area for tumor therapy, pharmacological inhibition of mitogenic signaling by targeting these molecules could be a challenge for pterygium treatment.”

Round 2

Reviewer 2 Report

The authors have improved the manuscript for publication.